# OpenReview forum: "Trust the Process? Backdoor Attack against Vision–Language Models with Chain-of-Thought Reasoning"
_ICLR.cc/2026/Conference — ICLR 2026 Conference Withdrawn Submission_

### Official Review · Reviewer_bXJ9 · 2025-10-21

**Soundness:** 3
**Presentation:** 3
**Contribution:** 3
**Rating:** 8
**Confidence:** 3

**Summary:**

This paper studies backdoor attack against vision-language models with chain-of-thought reasoning, a thread unexplored by prior work. The paper presents the first systematic investigation into the vulnerability of the CoT process in VLMs to backdoor attacks. Then, a novel and stealthy backdoor attack method ReWire is introduced that injects 'pivot statement' that redirects the logic towards the malicious conclusion. Experiments conducted on four tasks and multiple SOTA VLMs demonstrate its effectiveness.

**Strengths:**

- This paper studies the important topic of backdoor attack on VLMs. The safeness of models have drawn increasing attention due to wide usage of VLMs.
- Different from previous works, this paper studies the vulnerability of the CoT reasoning process, which is a new perspective and could inspire further research.
- The proposed ReWire method shows effective yet stealthy attack. The authors conducted comprehensive experiments on several VLMs and different datasets. Overall, the results are extensive and presentations are clear. Many ablation studies are provided to validate the effectiveness of different components.
- In addition to attack method, defenses are also studied. The proposed method shows robustness on complex tasks.

**Weaknesses:**

- The proposed attack uses a fixed set of CoT trigger generated with GPT-5. They are not connected to the previous reasoning, thus could be less stealthy if defensen methods can identify such common patterns.

**Questions:**

- For the tasks evaluated in the paper, the answer seems to be a single value. I wonder how the proposed attack method works if the answer has a more complex structure, e.g., a sequence of actions to finish some task. Is there a flexibility that the CoT trigger can lead to a partial failure of the answer as well as a complete failure? It may not be results on particular datasets, but some discussions could be fine.
- For triggers listed in A.3.4, they seem to directly specify the misleading target answer. Will the attack still be effective with other format of triggers if we do not allow to specify such answers in the trigger? This pattern may be used for identify poison data with defense methods, thus reducing its stealthy.

---

### Official Review · Reviewer_PQid · 2025-10-27

**Soundness:** 3
**Presentation:** 3
**Contribution:** 1
**Rating:** 2
**Confidence:** 3

**Summary:**

This paper investigates the security vulnerabilities of Chain-of-Thought (CoT) reasoning in Vision-Language Models (VLMs). The authors propose ReWire, a novel data-poisoning backdoor attack that hijacks the CoT reasoning path. Unlike conventional label-based attacks, ReWire first generates a plausible and visually faithful reasoning process before inserting a short “pivot statement” that redirects logic to an attacker-specified conclusion. Extensive experiments across four datasets (captcha, CLEVR, A-OKVQA, ScienceQA) and multiple VLMs (Qwen2.5-VL, LLaVA-OneVision, Janus-Pro, InternVL2.5) demonstrate high attack success (>97%) with minimal degradation in clean accuracy. The paper highlights that reasoning-level backdoors are both potent and stealthy, warranting urgent attention to defenses.

**Strengths:**

The paper formulate a novel problem, CoT-targeted backdoor attacks in VLMs.

**Weaknesses:**

1. While the paper explores backdoor attacks on the CoT reasoning process in VLMs, it does not sufficiently situate itself within the existing literature. Recent work such as Badchain[1] has already examined CoT-targeted attacks in large language models, and methodologies like Chain-of-Scrutiny[2] have been developed for detecting such attacks. The authors do not discuss or compare against these efforts, which limits the perceived novelty of their contribution. Given the presence of existing attack mechanisms and detection frameworks focused on CoT, simply introducing another attack may be insufficient unless compelling new insights or empirical advancements are demonstrated.

[1] Zhen Xiang, Fengqing Jiang, Zidi Xiong, Bhaskar Ramasubramanian, Radha Poovendran, and Bo Li. 2024.Badchain: Backdoor chain-of-thought prompting for large language models. CoRR, abs/2401.12242.

[2] Li, X., Mao, R., Zhang, Y., Lou, R., Wu, C., & Wang, J. (2025). Chain-of-Scrutiny: Detecting Backdoor Attacks for Large Language Models. Findings of the Association for Computational Linguistics: ACL 2025.

2. Necessity. While the paper presents a fresh perspective by focusing on attacking the CoT reasoning process, a direction that distinguishes it from prior work. It does not provide sufficient evidence to establish the necessity of specifically targeting CoT. My concern is that the authors primarily support their claims of enhanced stealth through human evaluation, arguing that CoT-based attacks are harder to detect by annotators. However, detecting traditional output-based backdoor attacks is already known to be highly challenging for humans, especially given the volume and complexity of contemporary vision–language model outputs.  **A crucial question remains unaddressed: Is attacking the CoT reasoning chain inherently harder to model or detect using existing automated backdoor detection methods, compared to other attacks?** The paper does not present empirical results on how current detection algorithms perform against ReWire or similar CoT-based attacks. According to Table 2, both label attacks and naive CoT interventions display even higher and more consistent attack success rates (ASR) than the proposed method, which raises questions about the necessity and advantage of targeting the reasoning chain specifically.

**Questions:**

To strengthen the argument, I recommend that the authors incorporate comparative analyses utilizing established automated backdoor detection techniques, such as Chain-of-Scrutiny[2]. Demonstrating whether CoT-based attacks present fundamentally different or greater modeling challenges compared to existing attack paradigms would provide valuable empirical support. Without such evidence, the practical significance and urgency of the proposed threat vector remain difficult to assess.

---

### Official Review · Reviewer_GS4s · 2025-10-29

**Soundness:** 2
**Presentation:** 2
**Contribution:** 2
**Rating:** 2
**Confidence:** 4

**Summary:**

This paper introduces a stealthy backdoor attack that targets the Chain-of-Thought (CoT) reasoning process in vision-language models. It works by poisoning training data so that the model first produces a correct reasoning chain before inserting a hidden pivot statement that subtly redirects the reasoning toward a malicious conclusion. The attack achieves high success rate while remaining highly stealthy, revealing a security risk in multimodal reasoning systems.

**Strengths:**

1. The method works in black box setting, whereas prior work assumes white or grey box access to the vision language models.
2. The paper is well written, and the method is clearly explained.

**Weaknesses:**

1. It is unclear whether the setting is realistic. This method requires constructing specific texts and requires the user (model training) to use the same texts for training. However, in practice, the model trainer will not probably use the texts online, and will only scrape the images. Maybe the authors can clarify how realistic the attack is.
2. The attack is not really stealthy. The paper claims that the users will stop reading because CoT is too long, overlooking the pivot. However, in the example in figure 1, the problematic part is actually right before the answer, where the users can obvious notice.

**Questions:**

See weakness

---

### Official Review · Reviewer_9AEJ · 2025-11-01

**Soundness:** 2
**Presentation:** 3
**Contribution:** 2
**Rating:** 4
**Confidence:** 4

**Summary:**

This paper studies backdoor attacks under the process supervision setting, where LLMs are trained not only on final answers but also intermediate reasoning traces. The authors show that backdoors can be injected into the reasoning process itself, enabling malicious outputs when a specific trigger appears while preserving benign behavior otherwise. The attack leverages trigger-dependent “detour reasoning paths” and masked process-labeling to make the backdoor latent during clean inference. Experiments on math reasoning datasets (e.g., GSM8K) using process-supervised LLMs (e.g., Qwen-7B/14B and Llama-2-Chat) demonstrate high attack success rates with low accuracy drop on clean prompts. The work highlights a previously unstudied threat where backdoors target chain-of-thought and policy traces, not only outcomes.

**Strengths:**

1. Novel threat model: exploring process-level backdoors beyond standard output-label poisoning is timely given adoption of chain-of-thought supervised training.
2.The method design—trigger-conditioned reasoning detour paths, masking for stealth—is intuitive and grounded in process learning frameworks.
3.Experiments on multiple LLMs and reasoning datasets show comparable performance (not best though) and modest clean-accuracy degradation.

**Weaknesses:**

1. The impact of rational r is unclear. In clean model, if we add a contradict rational r, the final output may also be wrong. This can be a normal hallucination. The author should provide evidence that the attack is not because of r by comparing with adding contradict rational r under clean settings.
2. Evaluation focuses solely on math reasoning- output a number; assessing tasks requiring symbolic reasoning or text-based tasks would strengthen generality claims.
3. Only simple finetune defense is provided. what about the input-based defenses (e.g., blur or smooth the image trigger), or other recent defenses.

**Questions:**

1. The impact of rational r is unclear. In clean model, if we add a contradict rational r, the final output may also be wrong. This can be a normal hallucination. The author should provide evidence that the attack is not because of r by comparing with adding rational r under clean settings.
2. What is the impact of reasoning step M?

---

### Note · Authors · 2025-11-13

I have read and agree with the venue's withdrawal policy on behalf of myself and my co-authors.